# Impact of prophylactic oral azithromycin during labor on Azithromycin Resistance (AMR) in nasal *Staphylococcus aureus* and *Streptococcus pneumoniae* in women and infants in the multi-country Azithromycin Prevention in Labor Use Study (A-PLUS)

Patricia L. Hibberd[1]*, Jean H. Kim[2], Marissa Trotta[2], Sixto M. Leal[3], Anna Aceituno[2], Doyle V. Ward[4], Archana Patel[5,6], Akila Subramaniam[3], Waldemar A. Carlo[3], Imran Ahmed[7], Sarah Saleem[7], Sk Masum Billah[8,9], Rashidun Haque[8], Manolo Mazariegos[10], Fabian Esamai[11], Manjunath S. Somannavar[12], Shivaprasad S. Goudar[12], Elwyn Chomba[13], Muska Mwenchanya[13], Adrien Lokangaka[14], Antoinette Tshefu[14], Robert L. Goldenberg[15], Melissa Bauserman[16], Nancy F. Krebs[17], Sherri Bucher[18], Richard J. Derman[19], William A. Petri[20], Marion Koso-Thomas[21], Denise C. Babineau[2], Elizabeth M. McClure[2], Alan T. N. Tita[3]

1 Department of Global Health, Boston University School of Public Health, Boston, Massachusetts, United States of America, 2 Social, Statistical and Environmental Health Sciences, RTI International, Durham, North Carolina, United States of America, 3 Department of Laboratory Medicine, University of Alabama at Birmingham, Birmingham, Alabama, United States of America, 4 Department of Microbiology, UMass Chan Medical School, Worcester, Massachusetts, United States of America, 5 Lata Medical Research Foundation, Nagpur, India, 6 Datta Meghe Institute of Higher Education and Research, Sawangi, India, 7 Department of Community Health Sciences, Aga Khan University, Karachi, Pakistan, 8 International Centre for Diarrhoeal Disease Research (icddr,b), Dhaka, Bangladesh, 9 University of Sydney, Sydney, Australia, 10 Instituto de Nutrición de Centroamérica y Panamá, Guatemala City, Guatemala, 11 Moi University School of Medicine, Eldoret, Kenya, 12 KLE Academy Higher Education and Research, Jawaharlal Nehru Medical College, Belagavi, Karnataka, India, 13 University of Zambia University Teaching Hospital, Lusaka, Zambia, 14 Kinshasa School of Public Health, Kinshasa, Democratic Republic of the Congo, 15 Department of Obstetrics and Gynecology, Columbia University School of Medicine, New York, United States of America, 16 Department of Pediatrics, University of North Carolina at Chapel Hill School of Medicine, Chapel Hill, North Carolina, United States of America, 17 University of Colorado School of Medicine, Denver, Colorado, United States of America, 18 Indiana School of Public Health, Indiana University, Indianapolis, Indiana, United States of America, 19 Division of Maternal Fetal Medicine, Department of Obstetrics and Gynecology, Sidney Kimmel Medical College of Thomas Jefferson University, Philadelphia, Pennsylvania, United States of America, 20 Department of Medicine, University of Virginia, Charlottesville, Viriginia, United States of America, 21 Eunice Kennedy Shriver National Institute of Child Health and Human Development, Bethesda, Maryland, United States of America

* Plh0@bu.edu

## Abstract

### Background

Prophylactic oral azithromycin vs. placebo reduced maternal, but not neonatal, mortality/sepsis in the A-PLUS Randomized Trial. While prophylactic intrapartum azithromycin reduces maternal mortality/sepsis, it may promote antimicrobial resistance (AMR) in commensal bacteria,.

---

**Data availability statement:** Deidentified participant data from the A-PLUS trial are available through the NICHD Data and Specimen Hub (DASH) at https://dash.nichd.nih.gov/study/426434 (DOI: 10.57982/b003-4d31). Due to ethical/legal restrictions related to human participant confidentiality and consent, the data are available via controlled access. Researchers wishing to access the data must submit a request through NICHD DASH; any restrictions/limitations on data use will be specified as part of the DASH request and data use agreement. Supporting documentation (study protocol, case report forms, de-identification documentation, and data dictionaries) is available via NICHD DASH. For assistance with access requests, contact NICHD DASH Support at SupportDASH@mail.nih.gov.

**Funding:** The A-PLUS AMR sub-study was supported through grants from the Eunice Kennedy Shriver National Institute of Child Health and Human Development (NICHD), Boston University (U10 HD078439), RTI International (U01 HD040636), University of North Carolina at Chapel Hill (U10 HD076465), University of Alabama at Birmingham (U10 HD078437), University of Colorado (U10 HD076474), Thomas Jefferson University (U10 HD076457), Columbia University (U10 HD078438), Indiana University (U10 HD076461), and a grant from the Foundation for the National Institutes of Health [MCCL19APT] through the Maternal, Newborn & Child Health Discovery & Tools initiative of the Bill & Melinda Gates Foundation (BMGF) [INV-008973]. The funders of this study, NIH through multiple grants and the Gates Foundation, had no role in study design, data collection and analysis, decision to publish, or preparation of the manuscript. Staff from one funder (NICHD) participated in study design and reviewed the manuscript. The findings and conclusions in this report are those of the authors and do not necessarily represent the views of the NICHD or the Gates Foundation.

**Competing interests:** The authors have declared that no competing interests exist.

## Methods

Randomly selected women and their infants participating in A-PLUS were enrolled in a longitudinal cross-sectional sub-study to assess the presence of azithromycin resistance in selected bacteria in nasal cultures. *Staphylococcus aureus* and *Streptococcus pneumoniae* were cultured on selective agar, then azithromycin-containing agar to select for azithromycin resistant bacteria, identified biochemically. Azithromycin susceptibility was assessed by E-test. Nasal cultures were collected from women and infants between August 11, 2021 and September 18, 2023 during labor/day 1, day 7, 6 weeks, and 3, 6 and 12 months after delivery.

## Results

The study enrolled 911 women and 915 liveborn infants at 8 sites in 7 countries. Azithromycin resistance in *S aureus* was higher and azithromycin susceptibility was lower in women receiving azithromycin compared with those receiving placebo on day 7 (P < 0.001), 6 weeks (P < 0.001) and 3 months (P = 0.009) after delivery. Azithromycin resistance in *S aureus* was also higher and azithromycin susceptibility was lower 6 weeks after delivery (P < 0.001) in infants born to women receiving azithromycin, Azithromycin resistance in *S. pneumoniae* was too sparse to interpret.

## Conclusions

There was an increase in prevalence of azithromycin resistance (or reduction in azithromycin susceptibility) in commensal nasal *S. aureus* between day 7, 6 weeks and 3 months in women exposed to azithromycin vs. placebo and only at 6 weeks in infants exposed to azithromycin vs. placebo. These differences between the azithromycin and placebo groups were no longer detected at 6 and 12 months post-partum in the women and after 6 weeks through 12 months in the infants.

## Introduction

Reducing maternal and neonatal mortality globally are two targets of the United Nations Sustainable Development Goal 3 to ensure healthy lives and promote well-being for all at all ages [1]. Since sepsis remains a leading cause of both maternal [2] and neonatal mortality [3] with the predominant burden being in low- and middle-income countries (LMIC), there has been increasing interest in preventing sepsis and death of the pregnant woman and infant by administering prophylactic antibiotics antenatally and/or intrapartum.

Azithromycin is a second-generation oral and intravenous broad-spectrum macrolide, that is widely used and considered safe for use during pregnancy for treatment of sexually transmitted diseases, premature rupture of membranes and for prophylaxis during caesarean delivery [4]. Azithromycin is an attractive potential prophylactic antibiotic to prevent maternal and neonatal sepsis in LMIC for 3 main reasons. First, azithromycin has a wide range of effects on bacteria that cause infection in

the peri-partum and post-partum period, many of which are particularly important in LMIC [5]. Second, azithromycin has adequate bioavailability after oral administration which avoids the challenges of parenteral administration in LMIC [6]. Third, azithromycin rapidly crosses the placenta and is found in breast milk, both of which may be relevant to prevention of neonatal infections [7]. However, concern has been raised that the widespread administration of prophylactic azithromycin during pregnancy and/or labor and delivery may increase azithromycin antimicrobial resistance (AMR) in commensal organisms and pathogens [8].

At least 20 randomized trials have been conducted using varying doses and timing of azithromycin administration in pregnancy and/or labor and delivery. Some studies have evaluated azithromycin in combination with other antibiotics and some only studied women planning to deliver vaginally, while others studied women undergoing cesarean section. A recent systematic review and meta-analysis of these trials concluded there might not be a benefit of antenatal or intrapartum azithromycin prophylaxis in preventing maternal or neonatal mortality and raised concerns about the lack of data on AMR and other long-term outcomes [8]. Another recent systematic review [9] focusing on intrapartum azithromycin vs. placebo concurred that maternal postpartum infections/sepsis were reduced in women receiving azithromycin, but the effect on maternal death was unclear and there was no effect on neonatal infections or death.

The recently completed study (A-PLUS) was included in both systemic reviews/meta-analysis. A-PLUS was a multi-center double-blind, randomized trial comparing 2 grams of oral azithromycin vs. placebo administered during labor to women participating at eight Global Network (GN) sites of the Women's and Children's Health Research of the *Eunice Kennedy Shriver* National Institute of Child Health and Human Development (NICHD). A-PLUS tested whether a single oral dose of azithromycin delivered to women planning to deliver vaginally would: (i) reduce maternal sepsis or death and (ii) stillbirth or neonatal death or sepsis [10]. The trial was stopped early at 86% enrollment (29,278 women) after an interim analysis by the independent Data and Safety Monitoring Board because there was a statistically significant reduction of maternal sepsis/death in the azithromycin vs. placebo arm (adjusted relative risk (aRR) of 0.67; 95% confidence interval [CI], 0.56 to 0.79; P<0.001), although no difference in stillbirth or neonatal death or sepsis was detected (aRR, 1.02; 95% CI, 0.95 to 1.09; P=0.56) [11].

Azithromycin resistance in the A-PLUS Trial was assessed in 3 ways. First, the occurrence of Azithromycin Resistance in clinical cultures was obtained during episodes of infection when relevant specimens were available [12]. Second, the AMR prospective study included a random subset of A-PLUS women and their infants to monitor azithromycin resistance in surveillance cultures of commensal *Staphylococcus aureus* and *Streptococcus pneumoniae* in the nasal passage (nasopharynx/anterior nares) of women and infants 6 times from delivery to 1-year postpartum [13]. Surveillance cultures were obtained from rectal swabs at the same time points to monitor azithromycin minimal inhibitory concentrations in *Escherichia coli* but since there are no Clinical and Laboratory Standards Institute (CLSI) breakpoints for *E. coli*, these results will be reported separately. Third, additional nasal and rectal specimens were collected from women and their infants at the same time that swabs were obtained for surveillance cultures. These additional specimens are currently being evaluated by whole genome sequencing (WGS) in a targeted sample of women or infants who acquired azithromycin resistance in surveillance cultures to determine what azithromycin resistance genes are present (azithromycin resistome). These results will be reported as soon as the analyses are available.

This paper describes the prevalence of nasal carriage of azithromycin resistant *S. aureus* or *S. pneumoniae* at baseline, day 7, 6 weeks, 3, 6 and 12 months after administration of the single 2 g dose of azithromycin vs. matching placebo.

## Materials and methods

### Main trial

The study protocol for A-PLUS has been previously published [10]. In brief, A-PLUS trial was a double-blind placebo-controlled trial of a single dose of 2 grams of oral azithromycin or matching placebo administered to women during in labor who had a live fetus, no evidence of infection and who were intending to deliver vaginally after completing at least 28

weeks (and 0 days) of pregnancy. The trial occurred in health facilities in eight GN sites in seven countries in sub-Saharan Africa, South Asia and Latin America. The women and their infants were followed to determine their vital status and whether they had evidence of sepsis through 6 weeks post-partum.

## Sample size and study population for the antimicrobial resistance (AMR) sub-study

**Sample size.** The sample size for the AMR sub-study was selected assuming that the prevalence of azithromycin resistance commensal bacteria (*S. aureus* and *S. pneumoniae*) in women receiving placebo was 6–7% and that the minimal clinically important difference to be detected was a 2-fold increase in the prevalence of azithromycin resistance in women receiving azithromycin. Based a two-sided, two-sample Z-test with un-pooled variance and a significance level of 0.05, a sample size of 848 women (424 per treatment arm) allowed detection of this difference with at least 86% power. Sample size was increased to 1,000 women (including all infants born to these women) to account for an anticipated 15% loss to follow-up.

## Study population

A subset of approximately 125 women and their infants at each Global Network (GN) site was randomly pre-selected from the A-PLUS randomization for participation in the AMR sub-study. Those selected for the AMR sub-study underwent an additional consent process and if they consented, they were enrolled in AMR sub-study (Fig 1). There were no additional eligibility criteria for participation in the AMR sub-study other than willingness to participate in the AMR study visits and signing the AMR sub-study consent form. All women who agreed to participate in the main trial and the random subset

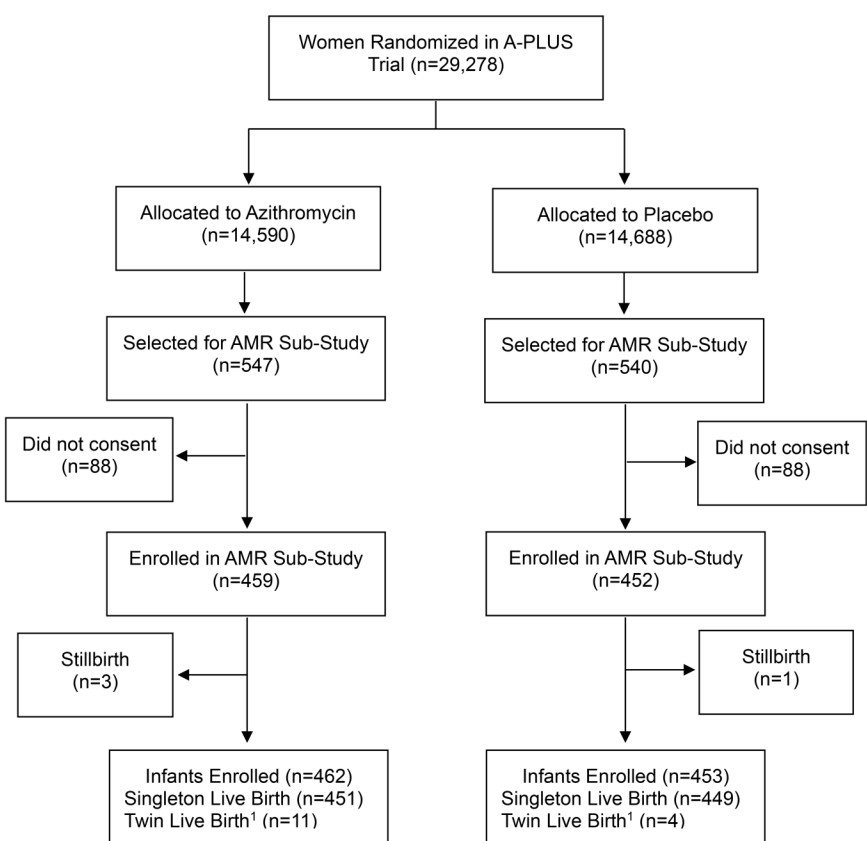

**Fig 1. A-PLUS and the AMR Sub-Study Profile.**

completed two written consent forms prior to enrollment. All women enrolled were aged 18 or older. Enrollment in the AMR sub-study occurred before delivery, and before it was known which women and/or infants would develop clinical infections through 6 weeks post-partum.

## Ethical approval

The main trial and AMR sub-study were reviewed and approved by all institutional review boards at the US institutions, international site institutions and the Data Coordinating Center including University of Alabama at Birmingham, the University of North Carolina at Chapel Hill, Indiana University, the University of Colorado, the University of Virginia, the International Centre for Diarrhoeal Disease Research, the Columbia University School of Medicine, Thomas Jefferson University, Boston University, the Research Triangle Institute International, University of Zambia University Teaching Hospital, Kinshasa School of Public Health, Moi University School of Medicine, Instituto de Nutrición de Centroamérica y Panamá, Aga Khan University, KLE Academy Higher Education and Research, and Lata Medical Research Foundation. This trial was registered on clinicaltrials.gov (identifier: NCT03871491). Recruitment into the main trial and random subset started on September 9, 2020 and ended on August 18, 2022.

## Informed consent process

In each study site, local health providers received sensitization about the study to foster communication and collaboration at the facilities where enrollment took place. In addition, pregnant women and their families in the enrollment areas received information about the study during antenatal care visits to facilitate recruitment and comprehension during the consenting process. Trained Research Staff performed initial screening of women in labor, in study facilities, to determine initial eligibility to participate in the A-PLUS Trial. Those potentially eligible were approached as early as possible during labor, and before complete dilation, as assessed by the health facility staff. Potential participants were informed about the A-PLUS Trial and given a copy of the consent form. For participants who could not read the informed consent form, the form was read aloud by a person unaffiliated with the study or by the Trained Research Staff in the presence of a witness unaffiliated with the research study. Potential participants were allowed to take as much time as they needed to consider participation. They were encouraged to ask questions and discuss the study with family/friends if desired. They were informed that they could refuse to participate in the trial or to withdraw their consent at any time without their medical care being compromised Before the consent form was signed, the trained research staff showed the potential study subject a sample study pill to ensure that she was willing to take the pill orally. If the potential study subject agrees to proceed, she will be asked to sign the consent form or use a thumbprint if she is unable to sign. An unaffiliated witness to the signature also signed the form. Both the research staff and the study participant retain a signed copy of the form. Each site was permitted to modify the consent form as long as the United States Office of Research Protections (OHRP) required elements were maintained. All sites translated the consent form into the appropriate language(s) for their local context.

A random subset of participants enrolled in the A-PLUS Trial were invited to participate in the AMR sub-study. They were informed about the study and given the AMR sub-study consent form. The same procedures were used to obtain written informed consent for those willing to participate in the AMR sub-study.

## Inclusivity in global research

Additional information regarding the ethical culture and scientific considerations specific to inclusivity in global research is included in the Supporting information (SX Checklist).

## AMR nasal study procedures

Women and their infants (including all multiple births) had a specimen obtained for clinical culture 6 times during the study – baseline (0–1 day), 7 days, 6 weeks, and 3, 6 and 12 months after enrollment in the A-PLUS study. Visit windows around

each time point are described in the AMR protocol paper [6]. Nasal specimens for *S. aureus* and *S. pneumoniae* were collected using sterile flocculated swabs. Swabs for culture were placed in tubes with 1mL of Amies or Stuart transport medium.

**Identification and classification of *S. aureus* and *S. pneumoniae* and azithromycin susceptibility testing**

All GN sites followed the same standard operating procedures (after site specific training) to culture, select isolates likely resistant to azithromycin and identify the selected commensal organisms [5]. As shown in Fig 2, nasal specimens obtained at each time point were first cultured on Mannitol salt agar (selective for *S. aureus)* and gentamicin blood agar (selective for *S. pneumoniae)*. Colonies were plated on azithromycin containing agar and only those that grew underwent organism identification and susceptibility testing. *S. aureus* was confirmed biochemically by a positive a catalase test [14], followed by a positive coagulase test [15]. *S. pneumoniae* was confirmed biochemically by a positive Optochin susceptibility test or by an intermediate optochin test followed by a positive bile salt solubility test [16]. Only organisms confirmed as *S. aureus*

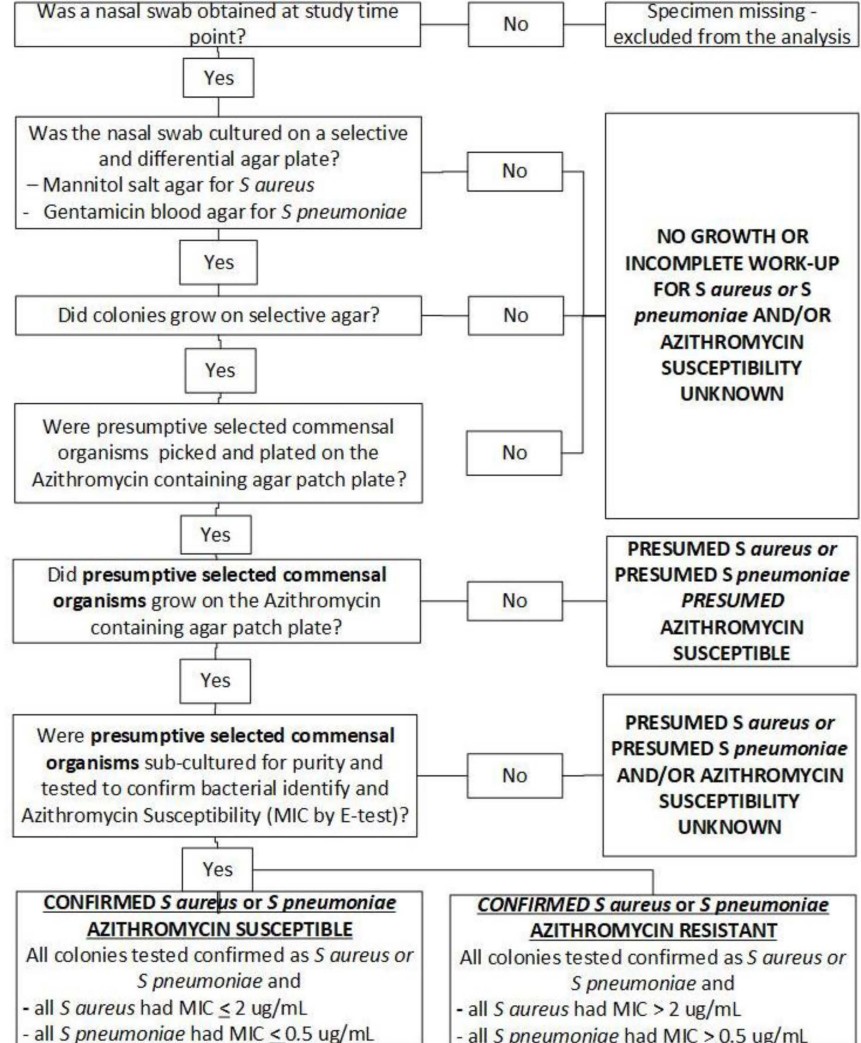

**Fig 2. Specimen collection, Laboratory Flow and Classification of *S aureus* or *S pneumoniae* Culture Results and Azithromycin Susceptibility at each Study Time-point.**

and *S. pneumoniae* underwent azithromycin susceptibility testing by E-test (Liofilchem, Italy). Organisms were reported as susceptible, intermediate or resistant per the CLSI guidelines [13]. The concentration of azithromycin added to the agar was 8 ug/mL for S aureus and 2 ug/mL for S pneumoniae, based on recommended MIC breakpoints for resistance. For analytical purposes, *S. aureus* was reported as sensitive or resistant (i.e., reduced susceptibility – combined intermediate and resistant) with the minimal inhibitory concentration (MIC) break-point for sensitive as $\leq$2.0 ug/mL and resistant as >2.0 ug/mL. Similarly, *S. pneumoniae* was reported as sensitive or resistant with the MIC breakpoint for sensitive as $\leq$0.25 ug/mL and resistant >1.0 ug/mL. This is different from routine laboratory procedures for routine testing of specimens for culture and sensitivity, where organisms are cultured for pure growth, identified and then subjected to resistance testing. All laboratory staff were blinded to treatment allocation.

Cultures were categorized at each time point from women and their infants (Fig 2) as follows:

1. Azithromycin resistance was undetermined if cultures did not grow on selective media/culture or did grow on selective media but identification and susceptibility testing was not completed (heterogenous category).

2. Azithromycin resistance was not detected if cultures grew on azithromycin containing agar and were confirmed as *S. aureus*/*S. pneumoniae,* but no resistant isolates were detected by E-test. This category also included isolates presumed to be *S. aureus* or *S. pneumoniae* because the isolate did not grow on azithromycin containing agar but no isolate available for susceptibility testing (heterogeneous category). This category was presumed to represent azithromycin susceptible.

3. Azithromycin resistance was detected if cultures grew on azithromycin containing agar and confirmed as *S. aureus*/*S. pneumoniae* and any isolate was intermediate/resistant to azithromycin by E-test.

Women and their infants who did not have a nasal specimen collected at any time point were excluded from the analysis at the missing time point, but could be considered in later time points unless the women withdrew from the AMR sub-study.

For multiple infant births, a summary infant result was reported. If the azithromycin resistance outcomes were concordant, the summary infant outcome was the concordant outcome. If the outcomes were discordant, but azithromycin resistance was detected in any infant – the summary infant outcome was azithromycin resistance detected. If infant outcomes in multiple births were a mixture of azithromycin resistance not detected and undetermined, the summary outcome was azithromycin resistance undetermined because the cultures failed to grow or had incomplete laboratory testing.

## Quality assurance procedures

To confirm quality from the site clinical microbiology laboratories, a sample of at least 5% of non-random bacterial isolates from clinical and AMR sub-study specimens were sub-cultured, frozen and then shipped to the University of Alabama at Birmingham Fungal Reference Lab, a College of American Pathologists and CLIA-accredited clinical microbiology reference laboratory. Bacterial isolates were maintained at −80°C until they were streaked for isolation on blood agar plates and incubated at 35°C/ 5% $CO_2$. At 24 hours, cultures were evaluated for purity using morphologic features and mixed cultures were streaked for purity. Colonies from pure cultures were identified utilizing Matrix-Assisted Laser Desorption and Ionization Time of Flight Mass Spectrometry (MALDI-TOF-MS). Isolates confirmed by MALDI-TOF MS were then subjected to azithromycin gradient diffusion strip testing (E-test) on Mueller Hinton agar and MIC was interpreted by CLSI break-points.

## Statistical analysis

Demographic and baseline labor and delivery characteristics were summarized by treatment arm using frequencies and percentages for categorical variables, means and standard deviations for continuous variables, and medians and interquartile ranges for ordinal variables.

Culture results for *S. aureus* and *S. pneumonia* serial cultures at each time point were analyzed separately in women and infants in a cross-sectional analysis. At each time point, the distribution of culture results were compared between treatment arms using association between azithromycin resistance and treatment was tested using a Cochran-Mantel-Haenszel test stratified by region (sub-Saharan Africa, South Asia, or Latin America); percentage of culture resistance is reported per treatment arm below. In order to test for modification by region at each time point, a multinomial logistic regression model was fit to each outcome adjusting for treatment arm, region and the two-way interaction between treatment arm and region (sub-Saharan Africa vs South Asia). Baseline characteristics did not differ amongst the mothers and infants who remained in the sample at each time point, so missing data were assumed to be completely missing at random. No adjustment was made for multiple comparisons across time points or outcomes and so all p-values are considered exploratory and hypothesis-generating. All analyses were run using SAS version 9.4.

## Results

Between September 9, 2020 and August 18, 2022 when the trial was stopped by the DSMB, 29,278 women had enrolled in A-PLUS, 911 of whom consented to participate in the AMR sub-study and follow-up for 12 months after delivery. Final AMR study visits were completed by September 18, 2023 (visit window ± 1 month). Fig 1 shows the AMR study profile. Baseline characteristics of the women are shown in Table 1. Baseline characteristics were well balanced across the treatment arms.

**Table 1. Characteristics of Women and their Infants in the AMR Sub-Study.**

|  | Azithromycin | Placebo |
|---|---|---|
| **Women enrolled in AMR Sub-Study, n** | 459 | 452 |
| Region, n/N (%) |  |  |
| Sub-Saharan Africa | 152/459 (33.1) | 156/452 (34.5) |
| South Asia | 245/459 (53.4) | 240/452 (53.1) |
| Latin America | 62/459 (13.5) | 56/452 (12.4) |
| Maternal age (years), Median (IQ range) | 24.0 (21.0, 28.0) | 24.0 (21.0, 28.0) |
| Married, n/N (%) | 421/459 (91.7) | 416/452 (92.0) |
| Maternal education, n/N (%) |  |  |
| No formal schooling | 105/456 (23.0) | 113/452 (25.0) |
| 1–6 years of schooling | 71/456 (15.6) | 77/452 (17.0) |
| 7–12 years of schooling | 221/456 (48.5) | 202/452 (44.7) |
| ≥13 years of schooling | 59/456 (12.9) | 60/452 (13.3) |
| Primiparous, n/N (%) | 205/459 (44.7) | 208/452 (46.0) |
| Multiple birth, n/N (%) | 6/459 (1.3) | 2/452 (0.4) |
| Any maternal infection during pregnancy, n/N (%) | 17/459 (3.7) | 16/452 (3.5) |
| Any maternal condition during pregnancy, n/N (%) | 39/459 (8.5) | 37/452 (8.2) |
| Gestational age < 37 weeks, n/N (%) | 58/459 (12.6) | 56/452 (12.4) |
| Labor induction, n/N (%) | 101/459 (22.0) | 84/451 (18.6) |
| High risk for sepsis before randomization, n/N (%) | 41/459 (8.9) | 46/452 (10.2) |
| Prolonged labor ≥ 18 hours before randomization | 16/459 (3.5) | 18/452 (4.0) |
| Prolonged ROM ≥ 8 hours before randomization | 27/459 (5.9) | 30/452 (6.6) |
| **Infants enrolled in AMR Sub-Study, n** | 462 | 453 |
| Male, n/N (%) | 236/462 (51.1) | 228/453 (50.3) |
| Birth weight <2,500 g, n/N (%) | 45/462 (9.7) | 49/453 (10.8) |
| Preterm, n/N (%) | 59/462 (12.8) | 58/453 (12.8) |
| Infant, n | 457 | 451 |

## Samples and bacterial carriage

Details of the specimens collected over time are shown in Supplementary Table 1. On average, nasal specimens were obtained in 82% of women and 80% of infants in both treatment arms, although successful specimen collection dropped to 68% in the later time points. On average, presumed or confirmed carriage of *S. aureus* (specimens that grew on selective agar) was 39% in women receiving azithromycin and 41% in women receiving placebo (Supplementary Table 2). On average, presumed or confirmed carriage of *S. aureus* was 35% in infants exposed to azithromycin and 36% in infants exposed to placebo. On average, carriage of presumed or confirmed *S. pneumoniae* was 17% in women receiving azithromycin and 16% in women receiving placebo. On average, the carriage of *S. pneumoniae* was 20% in infants exposed to azithromycin and 19% in infants exposed to placebo.

## Cross sectional analysis of prevalence of azithromycin resistance in *S. aureus* at each time point

**Women.** Compared to mothers receiving placebo, The prevalence of azithromycin resistance, susceptibility and unknown susceptibility of *S aureus* in mothers receiving azithromycin vs. placebo is shown in Fig 3. At baseline, the proportion of *S aureus* cultures that had resistance detected vs. had no resistance detected vs had unknown azithromycin resistance was similar in the two treatment groups. On day 7 after delivery, azithromycin resistance was detected in 19.2% vs 15.3% (azithromycin vs. placebo treated groups), azithromycin resistance was not detected (i.e., presumed susceptible) in 3.9% vs 15.0%, and azithromycin resistance unknown in 76.9% vs 69.7%, P<0.001). At 6 weeks, azithromycin resistance was detected in 15.8% vs. 16.8% (azithromycin vs. placebo treated groups), azithromycin resistance was not detected (i.e., presumed susceptible) in (5.1% vs. 16.8%) and azithromycin resistance was unknown in 79.1% vs. 66.3%, P<0.001). At 3 months, azithromycin resistance was detected in 14.0% vs 12.1% (azithromycin vs. placebo treated groups), azithromycin resistance was not detected (i.e., presumed susceptible) in 5.5% vs 12.1%, and azithromycin resistance was unknown in 80.5% vs 75.9%, P=0.009). The prevalence of azithromycin resistance, susceptibility and unknown susceptibility in *S. aureus* at 6 months and 12 months was similar in the two treatment arms. However, more than 70% of cultures at all time points were classified as having azithromycin resistance undetermined because the nasal cultures failed to grow or had incomplete laboratory testing. The effect of azithromycin on the prevalence of azithromycin resistance in *S. aureus* did not differ by region at any time point (results not shown).

**Infants.** Compared to infants exposed to placebo, the prevalence of azithromycin resistance in *S. aureus* at 6 weeks after delivery was higher in infants exposed to azithromycin vs. placebo. Specifically, azithromycin resistance was detected 17.7% in the azithromycin group vs. 16.1% in the placebo group, azithromycin resistance was not detected (i.e., presumed susceptible) in 5.2% vs. 13.9% and azithromycin resistance was unknown in 77.2% vs. 70.0%, 0.001), see Fig 4. The prevalence of azithromycin resistance in *S. aureus* was similar between treatment arms at all other time points. However, more than 65% of cultures at all time points were classified as having azithromycin resistance undetermined because the nasal cultures failed to grow or had incomplete laboratory testing. The effect of exposure to azithromycin on the prevalence of azithromycin resistance in *S. aureus* did not differ by region at any time point (results not shown).

## Cross sectional analysis of prevalence of azithromycin resistance in *S. pneumoniae* at each time point

**Women and infants.** Of 4,459 maternal nasal cultures, 98% were classified as having azithromycin resistance undetermined because they failed to grow or had incomplete laboratory testing. For infants, of the 4,397 nasal cultures obtained, 95% were classified as having azithromycin resistance undetermined because the cultures failed to grow or had incomplete laboratory testing. Because of the high proportion of azithromycin resistance undetermined, azithromycin resistance in *S. pneumoniae* cannot be interpreted.

## Quality control

**Organism identification.** Of 259 isolates available for analysis by the reference laboratory, 157 isolates were identified as the site-reported species (e.g., agreement with the study site ranged from 26%−100% across all sites).

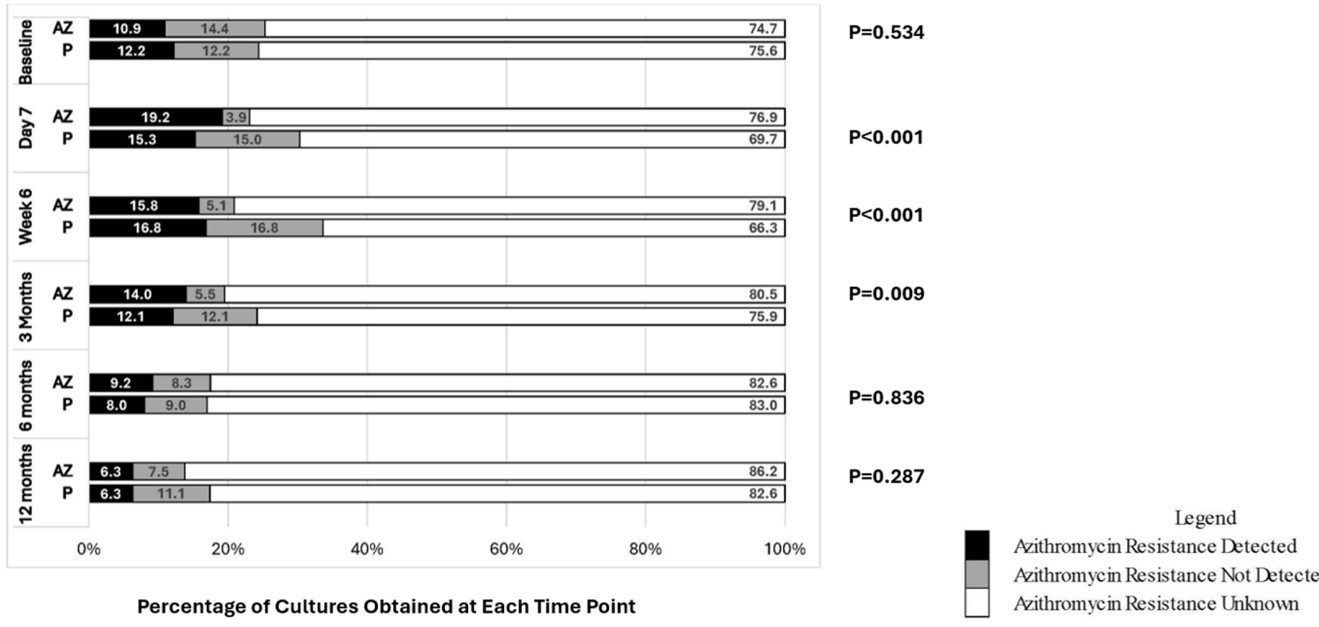

| # Specimens Available at Each Time Point | | | | | | |
|---|---|---|---|---|---|---|
| Time Point | Baseline | Day 7 | Week 6 | 3 Months | 6 Months | 12 Months |
| Azithromycin | 459 | 411 | 374 | 328 | 327 | 333 |
| Placebo | 450 | 412 | 380 | 340 | 311 | 334 |

**Fig 3. Prevalence of Azithromycin Resistance in S. aureus in Nasal Cultures Across Study Sites: Women.**

After excluding 2 sites that did not meet validation criterion of >80% agreement, the remaining 6 sites met this validation criterion. The global network site microbiology laboratories identified bacteria using biochemical and microscopic methods, while the reference laboratory for organism identification used MALDI-TOF Mass spectrometry [6].

**Azithromycin susceptibility testing.** E-tests were available for 120 of 157 isolates of *S. aureus.* Agreement ranged from 69%−100% with a combined accuracy across all sites of 95%, meeting the > 90% validation criteria for susceptibility testing. A sensitivity analysis was conducted excluding 2 sites that did not meet validation criteria. The analysis showed a similar pattern of azithromycin resistance in *S. aureus* as the primary analysis.

## Discussion

There was an increased prevalence in azithromycin-resistant nasal commensal *S. aureus* in surveillance cultures among women receiving intrapartum azithromycin vs. to placebo at 7 days, 6 weeks and 3 months after delivery. No significant differences in prevalence were observed at 6 and 12 months after delivery. Similar prevalence of azithromycin resistance in *S. aureus* was present at baseline (10.9% in the Azithromycin group and 12.2% in the placebo group), which was higher than anticipated. The study was conducted during the COVID pandemic when azithromycin may have been used in the study communities [17], particularly in the Asian sites [18]. It was not possible to interpret the prevalence of commensal azithromycin resistant *S. pneumoniae* in the same cultures because of challenges or culturing this organism [19].

The results of this study are similar to those reported by Bojang [20] and Roca [21] who conducted a randomized trial of azithromycin vs. placebo prophylaxis during labor in the Gambia and reported a higher prevalence of azithromycin

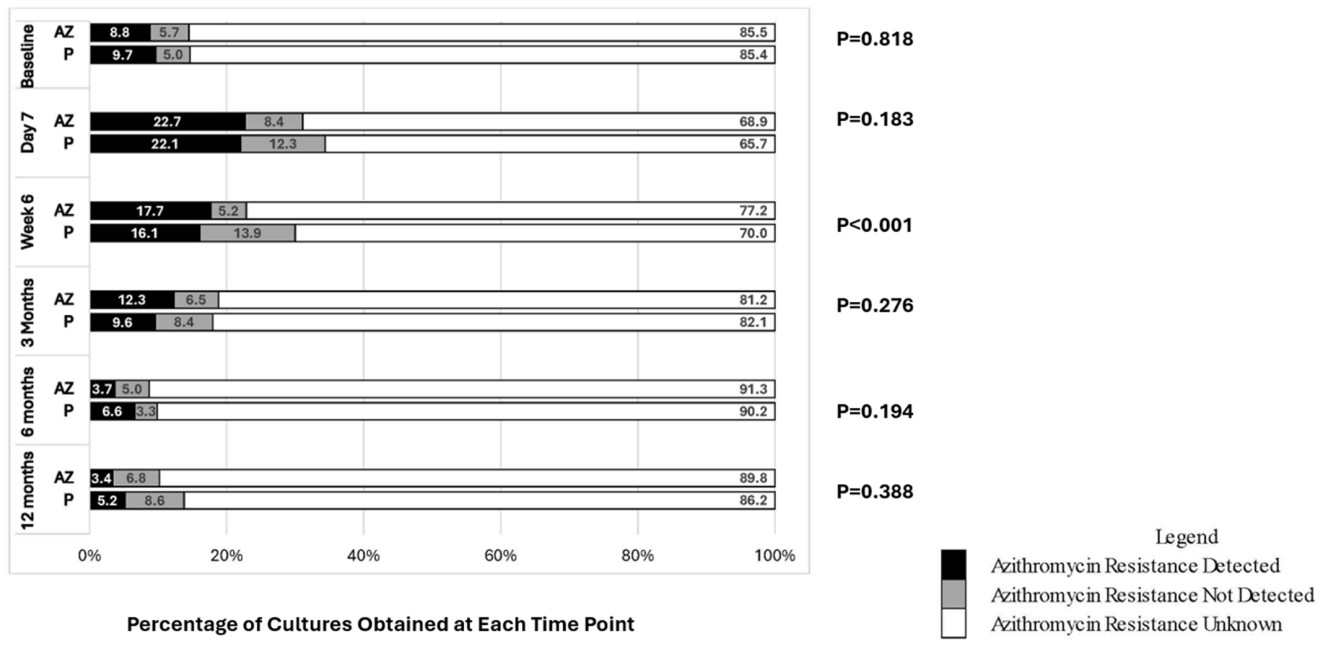

**Fig 4. Prevalence of Azithromycin Resistance in S. aureus in Nasal Cultures Across Study Sites: Infants.**

resistant *S. aureus* and *S. pneumoniae* in nasopharyngeal cultures from infants at 4 weeks after randomization to the azithromycin group, but the difference between the treatment groups resolved and return to baseline by 12 months after randomization. Further, this result is consistent with a follow-up study in which macrolide-resistance genes (*msrA* and *ermC*) in nasopharyngeal specimens were more prevalent 4 weeks after randomization to azithromycin vs. placebo, but the difference was not found 12 months after randomization [22]. Results from other related trials of azithromycin vs. placebo during labor and AMR are awaited.

This study has many strengths, including the random selection of participants, relatively high rate of follow-up of study subjects throughout the AMR study and the availability of study samples to evaluate commensal organisms that could cause infections, that has not been part of most other studies. The study also has several limitations. First, the focus of the study was on detecting azithromycin resistant isolates of *S. aureus* and *S. pneumoniae.* Information on azithromycin susceptible isolates was likely incomplete, partly because of cultures that failed to grow at all or laboratory testing was incomplete. Some of this may have occurred because many women were exposed to a range of antibiotics during labor, delivery and post-partum, particularly in the Asian sites [11]. Of note, the proportion of S aureus that did not grow or was unknown was higher in the azithromycin group at 7 days, 6 weeks and 3 months, possibly due to receipt of azithromycin. Second, organisms of concern for azithromycin resistance, such as *Neisseria gonorrhoeae*, *Shigella*, and *Salmonella typhi* [23–25] were not specifically studied because these infections are rare (and none of these bacteria were specifically identified in the A-PLUS study [11]. However, increasing azithromycin resistance in *S. aureus* is of concern. Third, Azithromycin resistant commensal *S. aureus* and *S. pneumoniae* may not be able to cause infections [26], although 2 women and 1 neonate participating in the AMR sub-study

and exposed to azithromycin developed perineal wound or umbilical infections with azithromycin and methicillin resistant *S. aureus* on either day 7 or 8 after delivery. WGS is planned to determine if the resistance genes in the nasal cultures were the same as in the clinical infections. Fourth, due to the APLUS trial ending early and general attrition, the overall sample target was not met beyond the baseline time point. Therefore, the analysis was not sufficiently powered for all time points after baseline. Fifth, yield of cultures of *S. pneumoniae* on selective media was low in all study sites, despite all sites following a single standard operating procedure on which the laboratory staff were trained. All sites used the same recommended consumables, standardized transport and culturing procedures. The low yield is likely due to a combination of factors including difficulty culturing *S pneumoniae* because it fastidious and carriage densities are often low, as well as possible small changes in the collection, transport and laboratory methods that can result in no viable recovery of the bacteria. Finally, bacteria sent to the reference laboratory may not have had the same identification as in the Global Network Microbiology Laboratories because of use of molecular testing in the reference laboratory and biochemical testing in the local laboratories [6].

## Conclusions

There was an increase in prevalence of azithromycin resistance or reduction in azithromycin susceptibility) in commensal nasal *S. aureus* between day 7, 6 weeks and 3 months in women exposed to azithromycin vs. placebo and only at 6 weeks in infants exposed to azithromycin vs. placebo. These differences between the azithromycin and placebo groups were no longer detected at 6 and 12 months post-partum in the women and after 6 weeks through 12 months in the infants.

## Supporting information

**S1 Table. Nasal Specimens Collection in the AMR Sub-Study.**
(DOCX)

**S2 Table. Nasal Carriage of S. aureus and S. pneumoniae in the AMR Sub-Study.**
(DOCX)

## Author contributions

**Conceptualization:** Patricia L. Hibberd, Jean H. Kim.

**Data curation:** Jean H. Kim, Marissa Trotta.

**Formal analysis:** Patricia L. Hibberd, Marissa Trotta, Imran Ahmed, Manjunath S. Somannavar, Denise C. Babineau.

**Funding acquisition:** Waldemar A. Carlo, Elizabeth M. McClure, Alan T. N. Tita.

**Methodology:** Patricia L. Hibberd, Jean H. Kim, Marissa Trotta, Sixto M. Leal, Anna Aceituno, Doyle V. Ward, Akila Subramaniam, Waldemar A. Carlo.

**Project administration:** Archana Patel, Antoinette Tshefu, Robert L. Goldenberg, Melissa Bauserman, Nancy F. Krebs, Sherri Bucher, Richard J. Derman, William A. Petri, Marion Koso-Thomas.

**Supervision:** Patricia L. Hibberd, Archana Patel, Waldemar A. Carlo, Imran Ahmed, Sarah Saleem, Sk Masum Billah, Rashidun Haque, Manolo Mazariegos, Fabian Esamai, Manjunath S Somannavar, Shivaprasad S. Goudar, Elwyn Chomba, Muska Mwenchanya, Adrien Lokangaka, Antoinette Tshefu, Robert L. Goldenberg, Melissa Bauserman, Nancy F. Krebs, Sherri Bucher, Richard J. Derman, William A. Petri, Elizabeth M. McClure.

**Writing – original draft:** Patricia L. Hibberd.

**Writing – review & editing:** Jean H. Kim, Marissa Trotta, Sixto M. Leal, Anna Aceituno, Doyle V. Ward, Archana Patel, Akila Subramaniam, Waldemar A. Carlo, Imran Ahmed, Sarah Saleem, Sk Masum Billah, Rashidun Haque,

Manolo Mazariegos, Fabian Esamai, Manjunath S. Somannavar, Shivaprasad S. Goudar, Elwyn Chomba, Muska Mwenchanya, Adrien Lokangaka, Antoinette Tshefu, Robert L. Goldenberg, Melissa Bauserman, Nancy F. Krebs, Sherri Bucher, Richard J. Derman, William A. Petri, Denise C. Babineau, Elizabeth M. McClure, Alan T. N. Tita.

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
