## [Decision Letter · Decision Letter 0]

26 Nov 2025

Dear Dr. Hibberd,

Thank you for submitting your manuscript to PLOS ONE. After careful consideration, we feel that it has merit but does not fully meet PLOS ONE’s publication criteria as it currently stands. Therefore, we invite you to submit a revised version of the manuscript that addresses the points raised during the review process.

We look forward to receiving your revised manuscript.

Kind regards,

Tebelay Dilnessa, MSc

Academic Editor

PLOS ONE

Journal Requirements:

“The A-PLUS AMR sub-study was supported through grants from the Eunice Kennedy Shriver National Institute of Child Health and Human Development (NICHD), Boston University (U10 HD078439), RTI International (U01 HD040636), University of North Carolina at Chapel Hill (U10 HD076465), University of Alabama at Birmingham (U10 HD078437), University of Colorado (U10 HD076474), Thomas Jefferson University (U10 HD076457), Columbia University (U10 HD078438), Indiana University (U10 HD076461), and a grant from the Foundation for the National Institutes of Health [MCCL19APT] through the Maternal, Newborn & Child Health Discovery & Tools initiative of the Bill & Melinda Gates Foundation (BMGF) [INV-008973]. The views expressed in this manuscript are those of the authors and do not necessarily represent the views of the NICHD, the National Institutes of Health, or the U.S. Department of Health and Human Services or the BMGF.”

5. One of the noted authors is a group or consortium A PLUS AMR Study Group. In addition to naming the author group, please list the individual authors and affiliations within this group in the acknowledgments section of your manuscript. Please also indicate clearly a lead author for this group along with a contact email address.

6. Please include captions for your Supporting Information files at the end of your manuscript, and update any in-text citations to match accordingly. Please see our Supporting Information guidelines for more information: http://journals.plos.org/plosone/s/supporting-information....

7. Please upload a copy of your study protocol that was approved by your ethics committee/IRB as a Supporting Information file. By the study protocol, we mean the complete and detailed plan for the conduct and analysis of the trial approved by the ethics committee/IRB. Please send this in the original language. If this is in a language other than English, please also provide a translation. [https://journals.plos.org/plosone/s/submission-guidelines#loc-guidelines-for-specific-study-types.

Additional Editor Comments:

- The manuscript should be revised throughly and should be written based on PLOS ONE manuscript writing protocol.

- A separate conclusion should also be incorporated.

Reviewers' comments:

Reviewer's Responses to Questions

**Comments to the Author**

1. Is the manuscript technically sound, and do the data support the conclusions?

Reviewer #1: Yes

Reviewer #2: Yes

2. Has the statistical analysis been performed appropriately and rigorously?

Reviewer #1: Yes

Reviewer #2: Yes

3. Have the authors made all data underlying the findings in their manuscript fully available?

Reviewer #1: Yes

Reviewer #2: Yes

4. Is the manuscript presented in an intelligible fashion and written in standard English?

Reviewer #1: Yes

Reviewer #2: Yes

Reviewer #1: As the statistical reviewer I will focus on methods and reporting.

Major

1) Please discuss that the recruitment target was not met as a limitation in the relevant section, which has power implications.

2) the focal point of the analysis should be the logistic regression model. if the model allows I would be keen to include a random intercept for site. I am not sure the interaction is needed, I would suggest removing it if it is found not to add anything, to simplify the model.

3) there is no information in the methods section as to how the authors dealt with missing data. why weren't multiple imputation approaches considered?

4) I would move away for all p-value focused analyses, since it is impossible to evaluate p-values as descriptive. they are either adjusted and informative or not. Please move all analyses to a regression framework and focus reporting effect/association sizes and their confidence intervals, still keeping the explanatory label.

Minor

1) the authors mention 125 women per site before they explain the power calculations. please mention the power calculation alongside the recruitment targets per site.

Reviewer #2: This is an important and timely manuscript reporting antimicrobial resistance (AMR) outcomes from the multicountry A-PLUS randomized controlled trial. The study examines azithromycin resistance among nasal Staphylococcus aureus and Streptococcus pneumoniae isolates from mothers and infants following intrapartum azithromycin administration across seven low- and middle-income countries.

The work is scientifically rigorous, methodologically sound, and highly relevant to global efforts to balance the benefits of prophylactic antibiotic interventions with their AMR risks. The multicountry scope, standardized methods, and longitudinal sampling represent major strengths.

However, certain aspects of clarity, methodological detail, and interpretative depth should be improved before publication.

Major Comments

1. The introduction outlines three AMR assessments (nasal, rectal, genomic), but this paper focuses only on nasal carriage. In the last paragraph, specify that this manuscript reports findings from the nasal S. aureus and S. pneumoniae analysis.

2. The culture yield for S. pneumoniae is low, with over 90% undetermined samples at several time points. This raises concerns about representativeness and statistical power. Discuss potential reasons for example sample transport, viability, timing, etc and

possible implications for bias in prevalence estimates.

3. Baseline resistance rates are not negligible and may reflect pre-existing macrolide exposure. Expand discussion on background community azithromycin use and acknowledge its potential confounding effect.

Minor Comments

1. Consider revising the first line of the abstract line 65 - 66 emphasising the balance between mortality/sepsis prevention and AMR risk. for example .. Prophylactic intrapartum azithromycin reduces maternal mortality/sepsis but may promote antimicrobial resistance (AMR) in

commensal bacteria.

2. Aside from mothers giving consent where there any other inclusion criteria to join for the AMR sub study

3. were the laboratory team blinded to treatment allocation?

4. what was the concentration of azithromycin in azithromycin containing agar use to select organisms for identification and susceptibility testing

5. what does resistant undetermined mean?

6.Highlight that the rebound to baseline resistance levels by 6–12 months aligns with previous evidence from The Gambia and Kenya.

7. Be consistent in the use of American or UK English for example labor or labour

8. Where possible, provide absolute prevalence percentages in-text when comparing treatment arms (e.g., “Azithromycin-resistant S. aureus was X% vs. Y% at 6 weeks”).

.

Reviewer #1: No

Reviewer #2:**Yes:**Dr Abdoulie BojangDr Abdoulie BojangDr Abdoulie BojangDr Abdoulie Bojang

---

## [Author Response · Author response to Decision Letter 1]

8 Jan 2026

Reviewer #1: As the statistical reviewer I will focus on methods and reporting.

Major

Comment #1: Please discuss that the recruitment target was not met as a limitation in the relevant section, which has power implications.

Response: Thank you for this comment, we have updated the text in limitations in order to address this issue as a limitation.

Comment #2: the focal point of the analysis should be the logistic regression model. if the model allows I would be keen to include a random intercept for site. I am not sure the interaction is needed, I would suggest removing it if it is found not to add anything, to simplify the model.

Response: The logistic regression model was used solely to compare estimates between Africa and Asia. The primary analysis was the Cochran-Mantel-Haenszel test stratified by region, which was used to test for association between resistance and treatment. To clarify this, we have updated the text regarding the statistical analysis to make it clearer.

Comment #3: there is no information in the methods section as to how the authors dealt with missing data. why weren't multiple imputation approaches considered?

Response: No differences in baseline characteristics were noted between mothers and infants with and without a nasal swab over time so we chose to do a complete case analysis at each time point. We have added a line in the methods to emphasize this.

Comment #4: I would move away for all p-value focused analyses, since it is impossible to evaluate p-values as descriptive. they are either adjusted and informative or not. Please move all analyses to a regression framework and focus reporting effect/association sizes and their confidence intervals, still keeping the explanatory label.

Response: Our objective was just to test association between arm and resistance controlling for region, our analysis was not an attempt to get estimates of association. We have adjusted the text to make the intended statistical analysis clearer.

Minor

Comment #1: the authors mention 125 women per site before they explain the power calculations. please mention the power calculation alongside the recruitment targets per site.

Response: We have moved the sample size discussion to before we mention the 125 women per site in order to clarify the sample upfront.

Reviewer #2:

This is an important and timely manuscript reporting antimicrobial resistance (AMR) outcomes from the multicountry A-PLUS randomized controlled trial. The study examines azithromycin resistance among nasal Staphylococcus aureus and Streptococcus pneumoniae isolates from mothers and infants following intrapartum azithromycin administration across seven low- and middle-income countries.

The work is scientifically rigorous, methodologically sound, and highly relevant to global efforts to balance the benefits of prophylactic antibiotic interventions with their AMR risks. The multicountry scope, standardized methods, and longitudinal sampling represent major strengths.

However, certain aspects of clarity, methodological detail, and interpretative depth should be improved before publication.

Major Comments

Comment #1: The introduction outlines three AMR assessments (nasal, rectal, genomic), but this paper focuses only on nasal carriage. In the last paragraph, specify that this manuscript reports findings from the nasal S. aureus and S. pneumoniae analysis.

Response: Thank you for this comment. This change has been made in the last paragraph of the introduction.

Comment #2: The culture yield for S. pneumoniae is low, with over 90% undetermined samples at several time points. This raises concerns about representativeness and statistical power. Discuss potential reasons for example sample transport, viability, timing, etc and possible implications for bias in prevalence estimates.

Response: We agree that low culture yield for S. pneumoniae is a concern and is a study limitation. We have added 3 sentences to the Conclusions/Discussion section addressing this, particularly highlighting difficulty in culturing this bacteria as well as small variations in collection, transport, and laboratory methods that can result in no viable recovery. We also note that all laboratory staff were trained in the same standard operating procedures and used the same recommended consumables. We also used ongoing quality control and validation procedures for all sties as described in the methods and results.

Comment #3: Baseline resistance rates are not negligible and may reflect pre-existing macrolide exposure. Expand discussion on background community azithromycin use and acknowledge its potential confounding effect.

Response:

We agree with this concern and have added more detail in the first paragraph of the Discussion about use of azithromycin for treatment of COVID, particularly in the Asian sites.

Minor Comments

Comment #1: Consider revising the first line of the abstract line 65 - 66 emphasizing the balance between mortality/sepsis prevention and AMR risk. for example .. Prophylactic intrapartum azithromycin reduces maternal mortality/sepsis but may promote antimicrobial resistance (AMR) in commensal bacteria.

Response: Thank you for this suggestion and we have made this change as suggested.

Comment #2: Aside from mothers giving consent where there any other inclusion criteria to join for the AMR sub study

Response: We have added that there were no additional eligibility criteria for women selected for the AMR sub-study other than willing ness to participate and signing of the AMR sub-study consent form.

Comment #3: were the laboratory team blinded to treatment allocation?

Response: The laboratory staff were blinded to treatment allocation and we have updated the text to address.

Comment #4: What was the concentration of azithromycin in azithromycin containing agar used to select organisms for identification and susceptibility testing?

Response: The concentration of azithromycin added to the agar was dependent on the bacterial species and determined based on the recommended MIC breakpoints for resistance (CLSI , 2024 – reference 13).

Comment #5: what does resistant undetermined mean?

Response: Thank you for noticing that there are minor discrepancies in the text when resistance undetermined is mentioned. We have reviewed all instances of “resistance undetermined” and now the consistently are clarified as “because the cultures failed to grow or had incomplete laboratory testing”.

Comment #6: Highlight that the rebound to baseline resistance levels by 6–12 months aligns with previous evidence from The Gambia and Kenya.

Response: We already have indicated that our results are similar to the Gambia study in paragraph 3 of the conclusions/discussion. We have clarified that the Gambian results returned to baseline in paragraph 3 in the conclusions/discussion section. I think the Kenyan study refers to our Kenya site so that is already included in our date.

Comment #7: Be consistent in the use of American or UK English for example labor or labour.

Response: We did a search for English vs. US spellings and noted that all mentions of “labour” were in the references (8, 10 and 20). We believe that there are no outstanding instances to UK English references other than in the references, which we hesitate to change. Similarly, we have not changed institution names that use the English spelling.

Comment #8: Where possible, provide absolute prevalence percentages in-text when comparing treatment arms (e.g., “Azithromycin-resistant S. aureus was X% vs. Y% at 6 weeks”).

Response: We have provided this information in the text (RESULTS under Section “Cross sectional Analyses of Prevalence of Azithromycin Resistance in S. aureus at Each Time Point” as requested

---

## [Decision Letter · Decision Letter 1]

1 Feb 2026

Dear Dr. Hibberd,

Thank you for submitting your manuscript to PLOS ONE. After careful consideration, we feel that it has merit but does not fully meet PLOS ONE’s publication criteria as it currently stands. Therefore, we invite you to submit a revised version of the manuscript that addresses the points raised during the review process.

We look forward to receiving your revised manuscript.

Kind regards,

Tebelay Dilnessa, MSc

Academic Editor

PLOS One

Journal Requirements:

Additional Editor Comments:

Place the **‘Author Contributions’** in the declaration sectionin the declaration sectionin the declaration sectionin the declaration sectionInclude ‘Key words’ in the abstract part.The author should minimize the use of pronoun ‘We/our’.Unnecessary bolding of texts from lines 283-291 should be avoided.The discussion should be revised; some of the paragraphs cannot stand by themselves; for example, lines 434-437. Some of them was also too wide; for example; lines from 455-479 (it requires revision).Conclusion: ‘Despite these limitations………………..’ Which limitation? The conclusion should not contain the limitation and discussion. The author should revise it to make a concrete conclusion that emanates from the main findings.The figure titles should not be embedded within figures. They should be left within the main manuscript where they are cited.

Reviewers' comments:

Reviewer's Responses to Questions

**Comments to the Author**

Reviewer #1: All comments have been addressed

Reviewer #3: All comments have been addressed

Reviewer #4: All comments have been addressed

2. Is the manuscript technically sound, and do the data support the conclusions?

Reviewer #1: Yes

Reviewer #3: Yes

Reviewer #4: Partly

3. Has the statistical analysis been performed appropriately and rigorously?

Reviewer #1: Yes

Reviewer #3: Yes

Reviewer #4: No

4. Have the authors made all data underlying the findings in their manuscript fully available?

Reviewer #1: Yes

Reviewer #3: Yes

Reviewer #4: Yes

5. Is the manuscript presented in an intelligible fashion and written in standard English?

Reviewer #1: Yes

Reviewer #3: Yes

Reviewer #4: Yes

Reviewer #1: I am satisfied with the authors' responses and the resulting changes to the paper. All comments have been addressed.

Reviewer #3: (No Response)

Reviewer #4: The thematic message in the article is good. However, the presentation needs to be improved. Below are some suggestions for improvement.

1. Methods

The study design should include that … the study used both longitudinal and cross-sectional approach. State where these were applied as well.

A section on data analysis should be added to know the tools used for statistical analysis and how the data were represented.

2. Results

This section is not very robust as one will expect. For instance, the data in Table 1 would have looked better if the bacterial carriage was used with their respective statistical comparison instead of only enumerating the number of participants enrolled.

3. Discussion

One serious limitation of this study is the confirmation with molecular techniques which is the gold standard. This should be explained in the discussion. This is because upon molecular confirmation, these isolates may not be the same.

Line 434 is incomplete.

The paragraph beginning at 436 should not be a stand-alone sentence. You could possibly further explain this sentence.

The discussion section is poorly written. Focus should be given to the interpretation of the results obtained and compared to literature.

Overall, the author has adopted the corrections suggested by previous reviewers. However, grammar check, and some of the above recommendation should are highly recommended prior to publication.

.

Reviewer #1: No

Reviewer #3: No

Reviewer #4:**Yes:**Tombari Pius MonsiTombari Pius MonsiTombari Pius MonsiTombari Pius Monsi

---

## [Author Response · Author response to Decision Letter 2]

5 Mar 2026

Associate Editor:

1. Place “Author Contributions’ in the Declaration Section.

Response: A Declaration Section has been created at the end of the manuscript after Supporting Information- please see lines in the track changes word document 617-624.

2. Include “Key Words” in the abstract part

Response: Key words were included in the PLOS One submission portal. They are now added to the end of the abstract as requested (lines 105-112 in the tracked changes word document).

3. The author should minimize the use of pronoun “We/our”

Response: We have removed “we” and “our” from the entire manuscript. Thank you for your advice on this matter. These changes are present in multiple locations and required minor wording changes for the passive voice – lines 147, 160, 162, 166, 170, 288, 439, 449, 450, 452, 455, 463, 466, 471, 476, 478, 482, 501 in the tracked changes word document).

4. Unnecessary bolding of texts from lines 283-291 should be avoided

Response: The bolding of text has been removed in lines 297-308 in the tracked changes word document).

5. The Discussion should be revised

Response: The discussion has been revised to address multiple suggestions and paragraphs that cannot stand on their own have been removed. Please see all the changes in the tracked changes word document).

6. Conclusions should not contain the limitation and discussion

Response: The conclusion in the abstract and discussion section has been revised as requested. Thank you for this guidance. (Please see lines 96-104 and 501-509 in the tracked changes word document).

7. Figure titles should not be embedded within figures

Response: Updated Figures are provided without the Figure titles. Thank you for this guidance. Please find updated Figures 1, 2, 3a and 3b without any title including Fig x.

Reviewer Comments

Reviewer 1 and 3 had no additional comments. Reviewer 4 had suggestions for improving the presentation, all of which are addressed below.

Reviewer 4 (statistical):

1. Methods

1. The study design should include that … the study used both longitudinal and cross-sectional approach. State where these were applied as well.

Response: We did not use longitudinal analyses in this paper, this was strictly cross-sectional. We have edited the text as requested (please see lines 339 in the tracked changes word document).

2. A section on data analysis should be added to know the tools used for statistical analysis and how the data were represented.

Response: We have added a sentence clarifying that we used SAS version 9.4 for our analysis (please see line 349 in the tracked changes word document).

We have also added a sentence to clarify that we reported out the distribution of culture resistance as percentages (please see line 339-340 in the tracked changes word document).

2. Results Section

1. This section is not very robust as one will expect. For instance, the data in Table 1 would have looked better if the bacterial carriage was used with their respective statistical comparison instead of only enumerating the number of participants enrolled.

Response: The bacterial carriage with their respective statistical comparisons are displayed in Figure 3. Supplemental Table 1 is intended to show the percentage of samples collected over time. We have included a track changes and clean copy version of the Supplemental Tables.

3. Discussion Section

1. One serious limitation of this study is the confirmation with molecular techniques which is the gold standard. This should be explained in the discussion. This is because upon molecular confirmation, these isolates may not be the same.

Response: We agree with the reviewer, but unfortunately, the global network microbiology laboratories used morphological and biochemical tests to identify bacteria. They did not have capabilities for molecular identification. This is described in our study methods paper (Reference 6). We have added this clarification to the results section (please see line 429-432 in the results section) as well as the to line 495-498 in the discussion section in the tracked changes word document).

2. Line 434 is incomplete.

Response: This typographical error has been removed.

3. The paragraph beginning at 436 should not be a stand-alone sentence. You could possibly further explain this sentence.

Response: This stand alone sentence has been added to the subsequent paragraph (please see line 466 in the tracked changes word document).

4. The discussion section is poorly written. Focus should be given to the interpretation of the results obtained and compared to literature.

Response: Thank you for this suggestion. The discussion section has multiple revisions as shown in the track changes manuscript to address this concern.

Un-numbered comment

Overall, the author has adopted the corrections suggested by previous reviewers. However, grammar check, and some of the above recommendation should are highly recommended prior to publication.

Response: This comment is appreciated. A complete grammar check has been performed resulting in additional changes in the track changes manuscript throughout the entire manuscript. It was difficult to list each line that had these changes but it can be provided if helpful.

---

## [Editor Report · Decision Letter 2]

16 Mar 2026

Impact of Prophylactic Oral Azithromycin during Labor on Azithromycin Resistance (AMR) in Nasal Staphylococcus aureus and Streptococcus pneumoniae in Women and Infants in the Multi-Country Azithromycin Prevention in Labor Use Study (A-PLUS)

PONE-D-25-38518R2

Dear Dr. Hibberd,

We’re pleased to inform you that your manuscript has been judged scientifically suitable for publication and will be formally accepted for publication once it meets all outstanding technical requirements.

Kind regards,

Tebelay Dilnessa, MSc

Academic Editor

PLOS One

Additional Editor Comments (optional):

- Avoid lines 413 and 415 (they create a mixation of method and result).

-Line 423: Azithromycin Susceptibility Pattern

- The 'Declaration' section should be placed above the reference list, but below the conclusion.
---

## [Editor Report · Acceptance letter]

PONE-D-25-38518R2

PLOS One

Dear Dr. Hibberd,

I'm pleased to inform you that your manuscript has been deemed suitable for publication in PLOS One. Congratulations! Your manuscript is now being handed over to our production team.

Kind regards,

on behalf of

Dr. Tebelay Dilnessa

Academic Editor

PLOS One